# Video Global Motion Compensation Based on Affine Inverse Transform Model

**DOI:** 10.3390/s23187750

**Published:** 2023-09-08

**Authors:** Nan Zhang, Weifeng Liu, Xingyu Xia

**Affiliations:** 1School of Electrical and Control Engineering, Shaanxi University of Science and Technology, Xi’an 710021, China; z13474108512@163.com; 2School of Automation, Hangzhou Dianzi University, Hangzhou 310018, China; sukxiaomi0331@163.com

**Keywords:** image processing, global motion compensation, feature point matching, affine transformation, target detection

## Abstract

Global motion greatly increases the number of false alarms for object detection in video sequences against dynamic backgrounds. Therefore, before detecting the target in the dynamic background, it is necessary to estimate and compensate the global motion to eliminate the influence of the global motion. In this paper, we use the SURF (speeded up robust features) algorithm combined with the MSAC (M-Estimate Sample Consensus) algorithm to process the video. The global motion of a video sequence is estimated according to the feature point matching pairs of adjacent frames of the video sequence and the global motion parameters of the video sequence under the dynamic background. On this basis, we propose an inverse transformation model of affine transformation, which acts on each adjacent frame of the video sequence in turn. The model compensates the global motion, and outputs a video sequence after global motion compensation from a specific view for object detection. Experimental results show that the algorithm proposed in this paper can accurately perform motion compensation on video sequences containing complex global motion, and the compensated video sequences achieve higher peak signal-to-noise ratio and better visual effects.

## 1. Introduction

At present, intelligent analysis technology has realized the detection, identification, tracking, and human behavior analysis of moving targets. It is widely used in the military, intelligent transportation, medicine, and other fields [1,2,3]. Among them, the detection of moving objects is the most basic and critical link. In a video sequence, according to the motion of the recording device itself, object detection is mainly divided into target detection in the static background and target detection in the dynamic background [4,5,6,7].

The motion of the background is usually caused by the change in the position of the recording device, which is called global motion [8,9]. The movement of the foreground is the movement of the moving object relative to the recording device, which is a local movement [10]. In this paper, the dynamic background refers to the global motion caused by the transformation of the camera position during video shooting. The moving target to be detected refers to the local motion caused by the target movement in the video sequence.

Target detection in the static background mainly includes the frame difference method [11], background difference method [12], etc. These methods are very effective in static backgrounds and are well established with high accuracy [13,14]. However, global motion makes object detection in dynamic backgrounds more complicated than that in static backgrounds. Most of the target detection and segmentation algorithms are suitable for static backgrounds, but cannot be effectively applied to dynamic backgrounds [15].

Moving object detection under a dynamic background [16,17] mainly includes optical flow method [18,19] and global motion compensation method [20,21]. The global motion compensation method first estimates the global motion and then analyzes and calculates the motion parameters for motion compensation. Thus, the problem of object detection in a dynamic background is transformed into object detection in a static background. For global motion estimation, it is first necessary to obtain global motion parameters. Usually, we use the method of extracting the feature points of two adjacent frames of images and matching them.

The motion vector between images is obtained first, and then the motion parameters are obtained by fitting the motion model [22]. In practical applications, motion vectors are generally obtained by matching feature points between images. Commonly used image feature point matching algorithms include SIFT (Scale-invariant feature transform) [23], ORB (Oriented FAST and Rotated BRIEF) [24] and SURF (Speed Up Robust Features) [25], etc.

The SIFT algorithm has scale invariance and can detect a large number of key points in the image for fast matching. However, since the SIFT algorithm does not consider the geometric constraints of the space, it leads to a high mismatch rate [26,27]. ORB combines the FAST feature point detection [28] with the BRIEF feature descriptor [29], and it has been further improved and optimized from their original implementation. SURF has improved compared to the SIFT algorithm. Through the combination of the Harris feature [30] and the overall image, the running speed of the program is greatly improved and the mismatch rate of feature points is reduced [31].

Although ORB takes less time than SURF, it provides lower matching rates in rotation and shearing scenarios of different strengths. Therefore, SURF is considered a suitable compromise between speed and performance. After the feature point matching pair of the image is obtained by the SURF algorithm, it is necessary to use the MSAC (M-Estimate Sample Consensus) algorithm [32] to further obtain the motion parameters contained in the feature point matching pair. And fit the affine transformation [33] model to complete the global motion estimation.

The traditional global motion compensation method is to use the adjacent two-frame images to conduct a global motion estimation to obtain the motion model. By performing model transformation on the previous frame image to predict the next frame image, a compensated frame image is obtained [34,35]. When the moving object moves slowly, the difference between the moving object and the background motion in two consecutive frames of images is small, making it susceptible to be easily mistaken for the background. The frame difference between the latter frame image and the compensated frame image makes the extracted moving object incomplete [36].

C. Song et al. [37] introduce a multiscale motion compensation network (MS-MCN) that works with a pyramid flow decoder to generate multiscale optical flows and perform motion-compensated prediction of the current frame from the previous frame. X. Liu et al. [38] propose a novel dynamic local filter network to perform implicit motion estimation and compensation by employing, via locally connected layers, sample-specific and position-specific dynamic local filters that are tailored to the target pixels. Son, Hyeongseok et al. [39] proposed a motion compensation network that combines the detail features with pre-computed structure features using a structure injection scheme, and then uses a feature matching-based motion compensation module to estimate the motion between the current and previous frames. Although the global motion compensation method based on deep learning has good effect, it is more costly in terms of time and space due to high algorithm complexity.

In this paper, we refer to the traditional idea of global compensation and make improvements. After the affine transformation model is obtained by performing global motion estimation on the adjacent front and rear frame images, an inverse transformation model of the affine transformation is established to act on the latter frame image. And sequentially output compensated images of consistent size with a specific view. The video sequence composed of compensated images can directly perform frame difference operations to extract moving targets.

We aim at the problem of the high false alarm rate of moving target detection in video sequences caused by global motion and start with the conversion of dynamic background to static background in video sequences. We propose a global motion compensation algorithm based on an affine inverse transform model: By combining the SURF algorithm with MSAC processing, we obtain an affine transformation matrix representing the global motion of the video sequence. On this basis, an inverse transformation model of affine transformation is proposed, which is used to compensate for the global motion. We also realize the conversion from a dynamic background to a static background in a video sequence. Finally, the moving target is detected by the frame difference method.

## 2. Problem Description

The video sequence *V* under the dynamic background is regarded as an ordered set composed of *n* frames of images:(1)$$\left\{\begin{array}{c} V_{n}\left(\Theta_{n}\right)=\left\{I_{1}\left(\theta_{1}\right),I_{2}\left(\theta_{2}\right),\cdots ,I_{k}\left(\theta_{k}\right),I_{k+1}\left(\theta_{k+1}\right)\cdots I_{n}\left(\theta_{n}\right)\right\}\\ I_{k}\left(\theta_{k}\right)=I_{kb}\left(\theta_{kb}\right)\cup I_{ko}\left(\theta_{ko}\right)\\ I_{ko}\left(\theta_{ko}\right)=I_{ko_{1}}\left(\theta_{ko_{1}}\right)\cup I_{ko_{2}}\left(\theta_{ko_{2}}\right)\cdots \cup I_{ko_{i}}\left(\theta_{koi}\right)\cdots \cup {I_{ko}}_{m}\left(\theta_{kom}\right) \end{array}\right.$$

In the Equation (Equation 1), $I_{k}\left(\theta_{k}\right)$ represents the image of the *k*th frame; $I_{kb}\left(\theta_{kb}\right)$ represents the set of pixels that make up the background in the image of the *k*th frame; $I_{ko}\left(\theta_{ko}\right)$ represents the pixel point set of all moving objects in the image of the *k*th frame. Assume that there are *m* moving targets in total, among which, $I_{ko_{i}}\left(\theta_{koi}\right)$ constitutes the pixel point set of the moving target marked as *i*.

The parameters for 2D transformations of an image encompass translation, rotation, scaling, shearing, mirroring, and composite transformations. Translation refers to the distance by which an image is moved along its horizontal and vertical directions. It is typically represented by horizontal translation value ($t_{x}$) and vertical translation value ($t_{y}$), which can be positive or negative. In rotation, we rotate the object at a particular angle θ from its origin. It is typically measured in degrees (°), where a positive value indicates counterclockwise rotation and a negative value indicates clockwise rotation. To change the size of an object, scaling transformation is used. In the scaling process, it either expands or compresses the dimensions of the object. Scaling can be achieved by multiplying the original coordinates of the object with the scaling factor to get the desired result.

Reflection is the mirror image of the original object. In other words, it can say that it is a rotation operation with 180°. In reflection transformation, the size of the object does not change. A transformation that slants the shape of an object is called the shear transformation. There are two shear transformations X-Shear and Y-Shear. One shifts X coordinates values and the other shifts Y coordinate values. In both cases, only one coordinate changes and the other preserves its values. However, transformations are often composite rather than existing in isolation. Composite transformations can be achieved by combining transformation matrices, as shown in Equation (Equation 2):(2)$$\left[T\right]\left[X\right]=\left[X\right]\left[T_{1}\right]\left[T_{2}\right]\cdots \left[T_{n}\right]$$
where $\left[T_{i}\right]$ Ti represents the transformation matrices for translation, rotation, scaling, mirroring, and shearing, respectively. [X] represents the video. The application scenario for the proposed algorithm in this paper is global motion compensation for videos captured by a camera. The degree of distortion in the videos is not high; hence, we set the transformation matrices for translation, rotation, and scaling.
(3)$$\left\{\begin{array}{c} \theta_{kb}={ [a_{k},b_{k},t_{k}]}^{T}\\ \theta_{ko_{i}}=\theta_{kb}+\eta_{ko_{i}} \end{array}\right.$$

In the Equation (Equation 3), $\theta_{kb}$ represents the motion state of the background pixel of the image in the *k*th frame, that is, the global motion state. The motion parameter $a_{k},b_{k},t_{k}$ describes the scaling, rotation, and translation state of the background, respectively; $\theta_{ko_{i}}$ represents the motion state of the foreground moving object based on the centroid of the moving object marked as I in the *k*th frame image. This motion state superimposes the current background motion state $\theta_{kb}$ and its own motion state $\eta_{ko_{i}}$.

Due to the superposition between the global motion and the motion of the moving object, it is difficult to distinguish between the moving object and the background, and the detection of the moving object is more difficult. In order to accurately and completely extract the moving target in the video sequence, it is necessary to remove the influence of the global motion on the target motion as much as possible. Then, the global motion compensation problem of video sequences under dynamic background can be described as follows: Find a suitable algorithm to obtain the parameters of the global motion estimator $a_{k}^{\prime },b_{k}^{\prime },t_{k}^{\prime }$. On this basis, seek a suitable global motion compensation method:(4)$$\left\{\begin{array}{c} {\hat{\theta }}_{kb}={ [a_{k}^{\prime },b_{k}^{\prime },t_{k}^{\prime }]}^{T}\\ \theta_{ko_{i}}^{\prime }=\theta_{kb}+\eta_{ko_{i}}-{\hat{\theta }}_{kb} \end{array}\right.$$

In the Equation (Equation 4), ${\hat{\theta }}_{kb}$ represents the estimation of the motion state of the background pixel of the image in the *k*th frame. That is the estimation of the global motion state of the image in the *k*th frame. The motion parameter $a_{k}^{\prime },b_{k}^{\prime },t_{k}^{\prime }$ describes the scaling, rotation, and translation states of the background in the global motion estimator, respectively. $\theta {{}^{\prime }}_{ko_{i}}$ represents the motion state of the moving object marked as I in the *k*th frame image after compensation.

## 3. Global Motion Compensation Algorithm Based on Affine Inverse Transform Model

### 3.1. Feature Point Matching

SURF is a commonly used feature point matching algorithm. When matching two images in the same scene, we first use the Hessian matrix to generate all the points of interest in the image. Then, we extract the feature points after eliminating the unreasonable interest points, and generate their descriptors. The matching pairs of feature points are obtained through the comparison of descriptors.

The Hessian matrix is a square matrix composed of the second-order partial derivatives of a multivariate function, which describes the gray gradient changes in all directions. It generates points of interest by obtaining all “suspicious” extremum points. Before constructing the Hessian matrix, the image needs to be Gaussian filtered, and the Gaussian blur coefficient is σ. Assume that a Hessian matrix *H* is established at a certain pixel in the grayscale image P(u,v) corresponding to the *k*th frame image:(5)$$H= [\begin{array}{cc} L_{uu}\left(P,\sigma \right) & L_{uv}\left(P,\sigma \right)\\ L_{uv}\left(P,\sigma \right) & L_{vv}\left(P,\sigma \right) \end{array}]$$

In the Equation (Equation 5), $L_{uu}\left(P,\sigma \right)$ is the convolution of the image $I_{k}\left(P\right)$ at the pixel point *P* and the second-order Gaussian template $\frac{\partial^{2}g\left(\sigma \right)}{\partial u^{2}}$, as shown in Equation (Equation 6). $L_{vv}\left(P,\sigma \right)$ and $L_{uv}\left(P,\sigma \right)$ are shown in the same way, as shown in Equations (Equation 7) and (Equation 8):(6)$$L_{uu}\left(P,\sigma \right)=\frac{\partial^{2}g\left(\sigma \right)}{\partial u^{2}}⊗I_{k}\left(P\right)$$
(7)$$L_{vv}\left(P,\sigma \right)=\frac{\partial^{2}g\left(\sigma \right)}{\partial v^{2}}⊗I_{k}\left(P\right)$$
(8)$$L_{uv}\left(P,\sigma \right)=\frac{\partial^{2}g\left(\sigma \right)}{\partial u\partial v}⊗I_{k}\left(P\right)$$

In which:(9)$$g\left(\sigma \right)=\frac{1}{2\pi \sigma^{2}}e^{-\left(u^{2}+v\right)/2\sigma^{2}}$$

It can be seen that the determinant of the Hessian matrix of each pixel is as follows:(10)$$\det(H)=L_{uu}L_{vv}-{\left(0.9L_{uv}\right)}^{2}$$

Equation (Equation 9) is also the discrimination of the Hessian matrix, where 0.9 is the weight coefficient. If the value of the determinant is not 0, it is determined that the pixel point is a possible extremum point, and this point is called an interesting point.

In order to obtain the feature points that can be used for feature matching in $I_{k}\left(\theta_{k}\right)$, the interest points need to be screened. At the d×d×d neighborhood of each interest point, we use a filter of size d×d to perform non-extreme value suppression: We compare each interest point with $d^{3}-1$ pixels in its scale space and 2D image space neighborhood. If it is not a maximum value or a minimum value, it will be eliminated, and the remaining key points will be saved as feature points. Figure 1 is a schematic diagram of non-extreme value suppression, assuming d=3. The interest points marked as × are compared to the surrounding 26 interest points marked as *O* to eliminate non-extremum points.

As shown in Figure 2, in order to obtain the feature point matching pair, the descriptor of the feature point needs to be generated, and the g×g block is established with each feature point as the center. Each block contains h×h pixels, and the block is then rotated to the feature orientation. Haar wavelet [40] is used to calculate the response value for each small block. Then, we use the feature vector shown in Equation (Equation 11) to represent the feature of the small block:
(11)$$F= [\sum d_{x},\sum \left|d_{x}\right|,\sum d_{y},\sum \left|d_{y}\right|]$$
where $\sum d_{x}$ and $\sum d_{y}$ represent the Haar wavelet response values in the horizontal direction and vertical direction relative to the characteristic direction, respectively. $\sum \left|d_{x}\right|$ and $\sum \left|d_{y}\right|$ represent the sum of the absolute values of the Haar wavelet responses in the horizontal direction and vertical direction relative to the characteristic direction, respectively. We combine the feature vectors of g×g blocks to obtain the z=4×g×g dimensional feature descriptor of the feature point.

According to the above method, the feature point sets of two adjacent frames of images $I_{k}\left(\theta_{k}\right)I_{k+1}\left(\theta_{k+1}\right)$ in the video sequence *V* under the dynamic background are respectively established: $\alpha^{k}=\left\{\alpha_{1}^{k},\alpha_{2}^{k},\cdots ,\alpha_{n_{k}}^{k}\right\}$, $\alpha^{k+1}=\left\{\alpha_{1}^{k+1},\alpha_{2}^{k+1},\cdots ,\alpha_{n_{k+1}}^{k+1}\right\}$. *E* is the number of feature points of $I_{k}\left(\theta_{k}\right)$. *F* is the number of feature points of $I_{k+1}\left(\theta_{k+1}\right)$. The descriptor sets corresponding to the feature point sets $\alpha^{k}$ and $\alpha^{k+1}$ are: $R\left(\alpha^{k}\right)=\left\{R\left(\alpha_{1}^{k}\right),R\left(\alpha_{2}^{k}\right),\cdots ,R\left(\alpha_{n_{k}}^{k}\right)\right\}$, $R\left(\alpha^{k+1}\right)=\left\{R\left(\alpha_{1}^{k+1}\right),R\left(\alpha_{2}^{k+1}\right),\cdots ,R\left(\alpha_{n_{k+1}}^{k+1}\right)\right\}$.

And the descriptors in the sets $R(\alpha^{k})$ and $R(\alpha^{k+1})$ all have z-dimensional features. By comparing the descriptors in the two descriptor sets, the matching of the image $I_{k}\left(\theta_{k}\right)$ and $I_{k+1}\left(\theta_{k+1}\right)$ feature points is completed. We use Euclidean distance to measure their similarity. The shorter the Euclidean distance, the better the matching degree of the two feature points. Finally, the best feature point matching point pair is selected as the feature point matching pair:(12)$$\left\{\begin{array}{c} R\left(\alpha_{p}^{k}\right)=\left(r_{p_{1}}^{k},r_{p_{2}}^{k},\cdots ,r_{p_{z}}^{k}\right)\\ R\left(\alpha_{p}^{k}\right)\in R\left(\alpha^{k}\right)\\ \alpha_{p}^{k}\in \alpha^{k}\\ p=1,2,\cdots ,n_{k} \end{array}\right.$$
(13)$$\left\{\begin{array}{c} R\left(\alpha_{q}^{k+1}\right)=\left(r_{q_{1}}^{k+1},r_{q_{2}}^{k+1},\cdots ,r_{q_{z}}^{k+1}\right)\\ R\left(\alpha_{q}^{k+1}\right)\in R\left(\alpha^{k+1}\right)\\ \alpha_{q}^{k+1}\in \alpha^{k+1}\\ q=1,2,\cdots ,n_{k+1} \end{array}\right.$$
(14)$$d\left(R\left(\alpha_{p}^{k}\right),R\left(\alpha_{q}^{k+1}\right)\right)={∥R\left(\alpha_{p}^{k}\right)-R\left(\alpha_{q}^{k+1}\right)∥}_{2}=\sqrt{\sum_{j=1}^{z}{\left(r_{pj}^{k}-r_{qj}^{k+1}\right)}^{2}}$$

Among Equations (Equation 12)–(Equation 14), $R(\alpha_{p}^{k})$ represents the descriptor of the *p* th feature point in the feature point description subset $R(\alpha^{k})$ of the image $I_{k}\left(\theta_{k}\right)$. $r_{p_{z}}^{k}$ represents the *z* th dimension feature of the descriptor. $R(\alpha_{q}^{k+1})$ represents the descriptor of the *q* th feature point in the feature point description subset $R(\alpha^{k+1})$ of image $I_{k+1}\left(\theta_{k+1}\right)$. $r_{q_{z}}^{k+1}$ represents the *z*th dimension feature of the descriptor.

$d(R\left(\alpha_{p}^{k}\right),R\left(\alpha_{q}^{k+1}\right))$ represents the measure of similarity between descriptors $R(\alpha_{p}^{k})$ and $R(\alpha_{q}^{k+1})$. When their value is less than the threshold λ, the feature points $\alpha_{p}^{k}$ and $\alpha_{q}^{k+1}$ corresponding to $R_{\alpha }\left(k\right)$ and $R_{\beta }\left(k+1\right)$ are called a feature point matching pair.

All feature point matching pairs that meet the above conditions are composed of feature point matching pairs between images $I_{k}\left(\theta_{k}\right)$ and $I_{k+1}\left(\theta_{k+1}\right)$:(15)$$S_{f}=\left\{\left(\alpha_{p}^{k},\alpha_{q}^{k+1}\right)\left|\alpha_{p}^{k}\in \alpha^{k},\alpha_{q}^{k+1}\in \alpha^{k+1},d\left(R\left(\alpha_{p}^{k}\right),R\left(\alpha_{q}^{k+1}\right)\right)\right.<\lambda \right\}$$

### 3.2. Global Motion Estimation

To obtain accurate motion parameters between adjacent frame images $I_{k}\left(\theta_{k}\right)$ and $I_{k+1}\left(\theta_{k+1}\right)$ in a video sequence *V* with dynamic backgrounds, and to estimate the background motion, it is necessary to eliminate a portion of feature point matches in $S_{f}$ that do not satisfy the motion transformation. In this paper, we combine the MSAC algorithm to remove outliers as much as possible from the set $S_{f}$ of feature point matches obtained by SURF. MSAC is a variant of RANSAC (Random Sample Consensus) [41] that overcomes the sensitivity to thresholds in RANSAC by modifying the cost function. Additionally, the MSAC algorithm not only considers the number of model data points but also reflects the degree of fit of the model data, making it superior to the RANSAC algorithm overall.

The majority of feature point matches in the set $S_{f}$ between image $I_{k}\left(\theta_{k}\right)$ and $I_{k+1}\left(\theta_{k+1}\right)$ can be generated by a single model, and there are at least $n_{s}$ point pairs ($n_{s}\leq \min\left(E,F\right)$) available for fitting the model parameters. These parameters are iteratively estimated as follows:

1. Randomly select $n_{k}$ feature points from the $S_{f}$ set and use them to fit a model $M_{k}$.

2. For the remaining feature points in $S_{f}$, calculate the transformation error for each point. If the error exceeds a threshold, mark it as an outlier; otherwise, identify it as an inlier and add it to the set IS for further record.

3. If the cost function *C* of the current inlier set IS is smaller than the cost function $C_{best}$ of the best inlier set $IS_{best}$, update $IS_{best}=IS$.

4. This entire process constitutes one iteration. If the number of iterations exceeds *k*, terminate the process. Otherwise, increment the iteration count and repeat the above steps. The value of the iteration count *G* is determined by Equation (Equation 16).
(16)$$G=\frac{\log(1-w)}{\log(1-\omega^{n_{s}})}$$
Whereas, *w* represents the probability of $n_{k}$ points being inliers after *G* iterations, typically set to 0.99; ω represents the ratio of inliers among $n_{k}$ feature points. The cost function during the iteration process is defined by Equation (Equation 17).
(17)$$C=\sum_{s\in S}L(W(s,\varphi ))$$
Whereas, *W* represents the error function, φ represents the estimated model parameters; *S* represents the set of matching point pairs; *s* represents a pair of matching points in the set; *L* represents the loss function, defined by Equation (Equation 18).
(18)$$L\left(\gamma \right)=\left\{\begin{array}{cc} e & \gamma \leq T\\ T & \gamma >T \end{array}\right.$$
In which, *e* represents the error that can be calculated using the error function *W*. *T* is the error threshold used to distinguish inliers. Furthermore, due to the inconsistent motion speeds between the foreground objects and the background, as well as the significantly lower number of feature points on the foreground objects compared to the background, during the iterative process of obtaining the final motion model, foreground feature points with transformation errors exceeding the threshold are identified as outliers and consequently eliminated.

The inlier point matches obtained through the MSAC algorithm between the images $I_{k}\left(\theta_{k}\right)$ and $I_{k+1}\left(\theta_{k+1}\right)$ are used to form a new set of matches, $S_{i}$. Additionally, the iterative process yields a global motion estimation model, an affine transformation matrix, for the motion between $I_{k}\left(\theta_{k}\right)$ and $I_{k+1}\left(\theta_{k+1}\right)$, as shown in Equation (Equation 18). Taking any pair of matching points, $\alpha_{p}^{k}\left(u_{p},v_{p}\right)$ and $\alpha_{q}^{k+1}\left(u_{q},v_{q}\right)$, from the set $S_{i}$, where $\alpha_{p}^{k}\in \alpha^{k}$ and $\alpha_{q}^{k+1}\in \alpha^{k+1}$, the affine transformation matrix $M_{k}$ is applied to the feature point $\alpha_{p}^{k}$ to obtain the point ${\hat{\alpha }}_{q}^{k+1}\left({\hat{u}}_{q},{\hat{v}}_{q}\right)$, as shown in Equation (Equation 19).
(19)$$M_{k}= [\begin{array}{ccc} a_{1}^{k} & b_{1}^{k} & t_{1}^{k}\\ b_{2}^{k} & a_{2}^{k} & t_{2}^{k}\\ 0 & 0 & 1 \end{array}]$$
(20)$$[\begin{array}{c} {\hat{u}}_{q}\\ {\hat{v}}_{q}\\ 1 \end{array}]= [\begin{array}{ccc} a_{1}^{k} & b_{1}^{k} & t_{1}^{k}\\ b_{2}^{k} & a_{2}^{k} & t_{2}^{k}\\ 0 & 0 & 1 \end{array}] [\begin{array}{c} u_{p}\\ v_{p}\\ 1 \end{array}]= [\begin{array}{c} a_{1}^{k}u_{p}+b_{1}^{k}v_{p}+t_{1}^{k}\\ b_{2}^{k}u_{p}+a_{2}^{k}v_{p}+t_{2}^{k}\\ 1 \end{array}]$$

In the Equation (Equation 20), $\left(t_{1}^{k},t_{2}^{k}\right)$ represents the translation of the two feature point coordinates, $\left(a_{1}^{k},a_{2}^{k}\right)$ reflects the corresponding rotational changes between the two feature points, and $\left(b_{1}^{k},b_{2}^{k}\right)$ reflects the corresponding scaling changes. These six parameters collectively serve as the estimation parameters ${\hat{\theta }}_{(k+1)b}$ for the global motion state of the image $I_{k+1}\left(\theta_{k+1}\right)$. After applying the aforementioned method to obtain the affine transformation matrices $M=\left\{M_{1},M_{2},\cdots ,M_{k},\cdots ,M_{n-1}\right\}$ that describe the global motion estimation between adjacent frames in the video sequence V with dynamic backgrounds, the changes in the motion parameters (a,b,t) can be used to estimate the variation in the background motion throughout the entire video sequence.

### 3.3. Global Motion Compensation

To describe the image, as shown in Figure 3, this paper defines a Cartesian coordinate system u−v with the top-left corner of the image as the origin. In this coordinate system, the coordinates (u,v) of each pixel represent the column and row numbers of that pixel in the image array, and the value corresponds to the grayscale intensity of the pixel. This u−v coordinate system is based on pixel units and serves as the image coordinate system.

However, since the image coordinate system alone cannot represent the physical position of each pixel in the image, this paper also establishes an x−y imaging plane coordinate system based on centimeters as the unit. This coordinate system has its origin at the center of the image. Figure 3 illustrates the relationship between the pixel coordinate system (u−v) and the imaging plane coordinate system (x−y).

Based on the obtained motion parameters between adjacent frame images, this paper proposes a global motion compensation algorithm based on the affine inverse transformation model. This algorithm compensates for the motion in the video sequence *V* with dynamic backgrounds on a frame-by-frame basis. To illustrate the process, let us consider the global motion compensation of the k+1th frame image $I_{k+1}\left(\theta_{k+1}\right)$.

First, the affine transformation matrices $\left\{M_{1},M_{2},\cdots ,M_{k}\right\}$ are inverted to obtain the inverse transformation matrices $\left\{M_{1}^{-1},M_{2}^{-1},\cdots ,M_{k}^{-1}\right\}$, where the affine transformation matrix $M_{k}$ is defined as shown in Equation (Equation 19). Next, the inverse transformation matrices are applied to each pixel of the image $I_{k+1}\left(\theta_{k+1}\right)$, as shown in Equation (Equation 21), resulting in a new frame image ${\tilde{I}}_{k+1}\left({\tilde{\theta }}_{k+1}\right)$.
(21)$$[\begin{array}{c} {\tilde{u}}_{k+1}\\ {\tilde{v}}_{k+1}\\ 1 \end{array}]=M_{1}^{-1}M_{2}^{-1}\cdots M_{k}^{-1} [\begin{array}{c} u_{k+1}\\ v_{k+1}\\ 1 \end{array}]$$
in which, $\left(u_{k+1},v_{k+1}\right)$ represents the pixel coordinates of $I_{k+1}\left(\theta_{k+1}\right)$, while $\left({\tilde{u}}_{k+1},{\tilde{v}}_{k+1}\right)$ represents the pixel coordinates of the resulting image ${\tilde{I}}_{k+1}\left({\tilde{\theta }}_{k+1}\right)$.

Due to the transformation calculations, the resulting pixel coordinates $\left({\tilde{u}}_{k+1},{\tilde{v}}_{k+1}\right)$ of $I_{k+1}\left(\theta_{k+1}\right)$ cause a change in the position of the image ${\tilde{I}}_{k+1}\left({\tilde{x}}_{k+1}\right)$ within its imaging plane. Additionally, each frame image has a different imaging plane coordinate system, resulting in variations in the imaging standards across the images. To address this issue, this paper adopts the imaging plane coordinate system $x_{1}-y_{1}$ of the first frame image $I_{1}\left(\theta_{1}\right)$ as the output view’s imaging plane coordinate system. The pixel coordinate range of $I_{1}\left(\theta_{1}\right)$ is also used as the pixel range of the output view, as shown in Figure 4. The image ${\tilde{I}}_{k+1}\left({\tilde{\theta }}_{k+1}\right)$ is transformed using the inverse affine transformation in relation to this output view, resulting in the globally motion-compensated image $I_{k+1}^{\prime }\left(\theta_{k+1}^{\prime }\right)$.

Since the original video sequence *V* contains global motion, different frames capture scenes that are not completely consistent. This means that certain pixels present in one frame may not appear in subsequent frames. Therefore, if the pixel coordinates of ${\tilde{I}}_{k+1}\left({\tilde{\theta }}_{k+1}\right)$ in the $x_{1}-y_{1}$ coordinate system exceed the pixel range of the output view (as illustrated by the non-shadowed region in Figure 4), it indicates that the corresponding scene composed of those pixels does not appear in the first frame image. As a result, those pixels are excluded from the output view. The final output image $I_{k+1}^{\prime }\left(\theta_{k+1}^{\prime }\right)$ only includes the pixels from ${\tilde{I}}_{k+1}\left({\tilde{\theta }}_{k+1}\right)$ that fall within the pixel range of the output view. As depicted in Figure 4, the pixel values in the shadowed region of the output view correspond to the corresponding pixel values in the image ${\tilde{I}}_{k+1}\left({\tilde{\theta }}_{k+1}\right)$, while the remaining region (non-shadowed region of the output view) is assigned a pixel value of 0.

The motion state $\theta_{(k+1)oi}^{\prime }$ of the motion target labeled as *i* in the k+1th frame image $I_{k+1}^{\prime }\left(\theta_{k+1}^{\prime }\right)$ can be represented by Equation (Equation 22).
(22)$$\theta_{\left(k+1\right)o_{i}}^{\prime }=\theta_{(k+1)b}+\eta_{\left(k+1\right)o_{i}}-{\hat{\theta }}_{(k+1)b}$$

In Equation (Equation 22), $\theta_{(k+1)b}$ represents the global motion state in the image $I_{k+1}\left(\theta_{k+1}\right)$, $\eta_{\left(k+1\right)o_{i}}$ represents the motion state of the motion target labeled as *i* in the image $I_{k+1}\left(\theta_{k+1}\right)$, and ${\hat{\theta }}_{(k+1)b}$ represents the estimation of the global motion state in the image $I_{k+1}\left(\theta_{k+1}\right)$.

The global motion compensation, using the inverse transformation model of the affine transformation, is applied to $\left\{I_{2}\left(\theta_{2}\right),I_{3}\left(\theta_{3}\right),\cdots ,I_{n}\left(\theta_{n}\right)\right\}$ consecutively. The resulting compensated output images are then concatenated in sequential order to form a new video sequence $V^{\prime }$.
(23)$$V_{n}^{\prime }\left(\Theta_{n}\right)=\left\{I_{1}\left(\theta_{1}\right),I_{2}^{\prime }\left(\theta_{2}^{\prime }\right),\cdots ,I_{k}^{\prime }\left(\theta_{k}^{\prime }\right),I_{k+1}^{\prime }\left(\theta_{k+1}^{\prime }\right),\cdots ,I_{n}^{\prime }\left(\theta_{n}^{\prime }\right)\right\}$$

In the sequence $\left\{I_{1}\left(\theta_{1}\right),I_{2}^{\prime }\left(\theta_{2}^{\prime }\right),\cdots ,I_{k}^{\prime }\left(\theta_{k}^{\prime }\right),I_{k+1}^{\prime }\left(\theta_{k+1}^{\prime }\right),\cdots ,I_{n}^{\prime }\left(\theta_{n}^{\prime }\right)\right\}$, the image coordinate system and imaging plane coordinate system are consistent throughout. They are established based on the reference image $I_{1}\left(\theta_{1}\right)$. The video sequence $V^{\prime }$ primarily captures the motion of foreground objects, while the background motion is almost negligible.

The adjacent frames of the video sequence $V^{\prime }$ are subtracted using frame differencing, resulting in the absolute difference of the grayscale values of the two frames, as shown in Equation (Equation 24). By comparing this difference with a threshold, the motion characteristics of the video can be analyzed to determine the presence of moving objects in the image sequence.
(24)$$D_{k}\left(\tau ,\upsilon \right)=\left|I_{k+1}^{\prime }\left(\tau ,\upsilon \right)-I_{k}^{\prime }\left(\tau ,\upsilon \right)\right|$$

In the Equation (Equation 24), $I_{k+1}^{\prime }\left(\tau ,\upsilon \right)$ and $I_{k}^{\prime }\left(\tau ,\upsilon \right)$ represent the grayscale values of the pixel points in the adjacent frames $I_{k}^{\prime }\left(\theta_{k}^{\prime }\right)$ and $I_{k+1}^{\prime }\left(\theta_{k+1}^{\prime }\right)$ of the video sequence $V^{\prime }$.

### 3.4. Computational Complexity

In SURF, assuming the image size is W×H and the sampling step is *s*, the complexity of feature detection is rough O((W/s)×(H/s)), where W/s and H/s represent the image size after sampling. For each detected interest point, the calculation of the main orientation involves computing Haar wavelet responses within a region and identifying the direction with the highest response. Assuming the complexity of Haar wavelet response calculation for each interest point is $O(P^{2})$, where *P* represents the size of the computation region, and if there are *N* detected interest points, the complexity of main orientation calculation is $O(N\times P^{2})$. With the descriptor dimension *D*, the number of wavelet responses *M*, the computation region size P×P, and *N* interest points, the complexity of descriptor generation is rough $O(N\times D\times M\times P^{2})$. In summary, the overall computational complexity of the SURF algorithm can be approximately represented as
(25)$$O\left(SURF\right)=O\left(\left(W/s\right)\times \left(H/s\right)\right)+O\left(N\times P^{2}\right)+O\left(N\times D\times M\times P^{2}\right)$$

The MSAC algorithm requires multiple iterations to find an appropriate model, where each iteration involves sampling, model estimation, and inlier-outlier classification. Assuming a total of *T* iterations, each iteration involves O(P) operations, where *P* is the number of data points. Therefore, the total complexity of iterations is O(T×P). In each iteration, MSAC randomly samples a subset of data from the dataset for model estimation. Assuming each sampling involves *S* data points, and a total of *T* iterations, the complexity of sampling is O(T×S). During each iteration, MSAC needs to estimate model parameters and compute the fitting error between data points and the model. The complexity of model estimation and evaluation typically depends on the problem’s characteristics and the chosen model. Assuming the complexity of model estimation and evaluation in each iteration is O(F), the total complexity for model estimation and evaluation is O(T×F). To sum up, the overall computational complexity of the MSAC algorithm can be approximately represented as
(26)O(MSAC)=O(T×P)+O(T×S)+O(T×F)
where *T* is the number of iterations, *N* is the number of data points, *S* is the sampling size, and *P* is the complexity of model estimation and evaluation.

In the global motion compensation module, the inversion of an affine transformation matrix has a computational complexity that can be considered at a constant level, denoted as O(1). Assuming that the computational complexity of matrix multiplication and vector addition is O(3), then the computational complexity of motion compensation is O(3).

The overall complexity of the algorithm in this paper is
(27)O(OURS)=O(SURF)+O(MSAC)+O(3)

Son et al. [39] present an efficient multi-task network (RVDMC) for real-time video deblurring and motion compensation that shares computation and features between tasks by extracting useful details and injecting structural information, enabling state-of-the-art efficiency and flexible quality-speed trade-offs. The complexity of this algorithm can be expressed as follows:

The network architecture of the algorithm is composed of several multi-task units (MTUs). Each MTU consists of three main components: multi-task detail network $F_{n}$, deblurring network $D_{n}$, and motion compensation network $M_{n}$, where *n* is the index of a multi-task unit. H, W is image height and width; C is the number of channels in feature maps; K is the kernel size of convolutions; S is the stride of convolutions; D is the size of the matching window for motion estimation; N is the number of multi-task unit stacks. Thus, the complexity for a single stack of $F_{n}$ is
(28)$$O\left(F_{n}\right)=O\left(H\times W\times C^{2}\times K^{2}\right)+N\times O\left(H\times W\times C^{2}\times K^{2}\right)=O\left(N\times H\times W\times C^{2}\times K^{2}\right)$$

In the motion compensation network $M_{n}$, only cost volume calculation is considered.
(29)$$O\left(M_{n}\right)=O\left(H\times W\times C^{2}\times K^{2}\right)$$

For deblurring network $D_{n}$, there is only a single convolution. The complexity of $D_{n}$ is:(30)$$O\left(D_{n}\right)=O\left(H\times W\times C^{2}\times K^{2}\right)$$

The total complexity of the RVDMC algorithm is:(31)$$O\left(RVDMC\right)=O\left(N\times H\times W\times C^{2}\times K^{2}\right)$$

As can be seen from Equations (Equation 27) and (Equation 31), the complexity of the proposed algorithm in this paper is much lower than the global motion compensation algorithm based on deep learning.

## 4. Experiment

In this section, we conducted background motion compensation experiments on two video sequences, $V_{P}$ and $V_{Q}$, with dynamic backgrounds. The $V_{P}$ sequence consists of 280 frames, while the $V_{Q}$ sequence consists of 210 frames.
(32)$$\left\{\begin{array}{c} V_{P\{1:280\}}\left(\theta_{p\{1:280\}}\right)=\left\{I_{P1}\left(\theta_{P1}\right),I_{P2}\left(\theta_{P2}\right),\cdots ,I_{P280}\left(\theta_{P280}\right)\right\}\\ V_{Q\{1:210\}}\left(\theta_{Q\{1:210\}}\right)=\left\{I_{Q1}\left(\theta_{Q_{1}}\right),I_{Q2}\left(\theta_{Q2}\right),\cdots ,I_{Q210}\left(\theta_{Q210}\right)\right\} \end{array}\right.$$

In experiments, the proposed algorithm is used to estimate and compensate for the motion of $V_{P}$ and $V_{Q}$ video sequences. Furthermore, we performed motion object detection on these sequences to demonstrate the effectiveness and accuracy of the proposed algorithm in background motion compensation.

### 4.1. Obtaining Valid Feature Point Matches

The video sequences $V_{P}$ and $V_{Q}$, chosen for the experiment, have frame sizes of the common display standard 1280×720. To highlight the effects of feature point matching, we selected two frames with a significant time interval for experimentation. Specifically, we chose the first frame and the 200th frame. We applied the SURF feature point extraction and matching techniques to these frames and combined the results with the MSAC algorithm to obtain inlier point matches. The resulting feature points and inlier point matches for $V_{P}$ are shown in Figure 5, while the results for $V_{Q}$ are displayed in Figure 6.

In Figure 5, there are a total of 1103 feature point matches and 619 inlier point matches between the first frame and the 200th frame of $V_{P}$. In Figure 6, there are 593 feature point matches and 155 inlier point matches between the first frame and the 200th frame of $V_{Q}$. It can be observed that the number of inlier points is significantly lower compared to the total number of feature points. This indicates that a portion of feature points that are not suitable for fitting the motion model has been eliminated.

### 4.2. Motion Estimation on Video Sequences

The affine transformation model consists of six motion parameters, as shown in Equation (Equation 19). Table 1 presents the values of these six parameters for the two sets of images mentioned in Section 4.1.

According to the affine transformation model, $a_{1}$ and $a_{2}$ represent the scaling of the recording device, with values close to 1 indicating minimal changes in image size. $b_{1}$ and $b_{2}$ represent the rotation of the recording device, with small values suggesting that the images have undergone little to no rotation. $t_{1}$ and $t_{2}$ represent the translation motion of the recording device, with larger values indicating significant translational movement of the device.

For all the consecutive frame images in the video sequences $V_{P}$ and $V_{Q}$, we followed the aforementioned steps and obtained a total of 279 and 209 sets of parameter values, respectively. These parameter values can be categorized into scaling parameters, rotation parameters, and translation parameters based on their nature. Additionally, corresponding two-dimensional line graphs were created to visualize the dynamic changes of the background motion parameters. The resulting plot depicting the variation of the background motion parameters is shown in Figure 7.

In Figure 7a,d, the scaling changes in the video sequences are reflected, b and e depict the rotation variations, while c and f illustrate the translation changes. Each of the six plots demonstrates different degrees of background motion variations, providing a comprehensive motion estimation analysis for the two video sequences. The motion estimation results reveal that both video sequences, $V_{P}$ and $V_{Q}$, exhibit background motion consisting of scaling, rotation, and translation transformations, without following any specific pattern or regularity.

### 4.3. Global Motion Compensation and Object Detection

After estimating the global motion, we apply the proposed background motion compensation algorithm to the video sequences $V_{P}$ and $V_{Q}$ with dynamic backgrounds. We perform an inverse transformation on the affine transformation matrices between adjacent frames in the video sequences $V_{P}$ and $V_{Q}$. Then, we apply the inverse transformation matrices to the subsequent frames, as shown in Equation (Equation 21). Finally, we output the new video sequences $V_{P}^{\prime }$ and $V_{Q}^{\prime }$ in the views created based on the standards $I_{P1}\left(\theta_{{}_{P1}}\right)$ and $I_{Q1}\left(\theta_{Q_{1}}\right)$, respectively.
(33)$$\left\{\begin{array}{c} {V^{\prime }}_{P\{1:280\}}\left({\theta^{\prime }}_{p\{1:280\}}\right)=\left\{I_{P1}\left(\theta_{P1}\right),I_{P2}^{\prime }\left(\theta_{P2}^{\prime }\right),\cdots ,I_{P280}^{\prime }\left(\theta_{P280}^{\prime }\right)\right\}\\ {V^{\prime }}_{Q\{1:210\}}\left({\theta^{\prime }}_{Q\{1:210\}}\right)=\left\{I_{Q1}\left(\theta_{Q1}\right),I_{Q2}^{\prime }\left(\theta_{Q2}^{\prime }\right),\cdots ,I_{Q210}^{\prime }\left(\theta_{Q210}^{\prime }\right)\right\} \end{array}\right.$$

The video sequences $V_{P}^{\prime }$ and $V_{Q}^{\prime }$ are the result of global motion compensation using the inverse affine transformation model, transforming them into video sequences with static backgrounds. Figure 8 shows the first frame image P1 in $V_{P}^{\prime }$ and the adjacent frames $I_{P125}^{\prime }\left(\theta_{P125}^{\prime }\right)$ and $I_{P126}^{\prime }\left(\theta_{P126}^{\prime }\right)$ after global motion compensation. Figure 9 shows the first frame image $I_{Q1}\left(\theta_{Q1}\right)$ in $V_{Q}^{\prime }$ and the adjacent frames $I_{Q41}^{\prime }\left(\theta_{Q41}^{\prime }\right)$ and $I_{Q42}^{\prime }\left(\theta_{Q42}^{\prime }\right)$ after compensation.

From Figure 8 and Figure 9, it can be observed that due to the presence of global motion in $V_{P}^{\prime }$ and $V_{Q}^{\prime }$, different frames capture slightly different scenes. After applying the proposed algorithm for global motion compensation, there may be pixels that fall outside the view boundaries or do not satisfy the output view criteria. These pixels result in varying degrees of black borders, as expected in the compensated images.

Based on the obtained video sequences $V_{P}^{\prime }$ and $V_{Q}^{\prime }$ with static backgrounds, motion detection can be performed by computing the frame differences between consecutive frames. To demonstrate the effectiveness of the algorithm proposed in the paper, a comparison is made between the direct frame difference images, the frame difference images obtained using a traditional global motion estimation algorithm with compensation frames, and the frame difference images obtained using the algorithm proposed in the paper. Figure 10 and Figure 11 illustrate this comparison.

Figure 10a,b represent the 223rd and 224th frames of the $V_{P}$ sequence, respectively. Figure 10c shows the frame difference image obtained using a traditional algorithm with compensation frames. Figure 10e shows the frame difference image obtained using the algorithm proposed in the paper.

By comparing the two frame difference images, significant differences in the object detection results can be observed. In Figure 10c, the direct frame difference result, the trees and buildings in the background are detected along with the moving objects. In Figure 10d, the frame difference result obtained using the traditional algorithm, although some moving objects are detected, the detection is incomplete. However, in Figure 10e, the result obtained using the algorithm proposed in the paper, the moving objects are detected, and their outlines are clearly visible.

Figure 11a,b represent the 134th and 135th frames of the $V_{Q}$ sequence, respectively. c shows the frame difference image obtained directly from these two frames. d shows the frame difference image obtained using a traditional algorithm with compensation frames. e shows the frame difference image obtained using the algorithm proposed in the paper.

In comparison to the $V_{P}$ sequence, the $V_{Q}$ sequence has more background motion, which leads to stronger interference in the direct frame difference result. In Figure 11c, the objects and the background are merged together, making it difficult to distinguish the moving objects. In Figure 11d, the result obtained using the traditional algorithm, there are more noise points and the edges of the objects are not clear. However, in Figure 11e, the result obtained using the algorithm proposed in the paper, the interference from the global motion is significantly reduced, and the moving objects are clearly detected.

In this paper, we adopt an objective quality metric called Peak Signal to Noise Ratio (PSNR) to demonstrate the effectiveness of the proposed algorithm. A higher PSNR value indicates lower image distortion. We calculate the PSNR values between the grayscale frames of adjacent frames in the dynamic background video sequences $V_{P}$ and $V_{Q}$. We also calculate the PSNR values between the compensated frames obtained by the traditional global motion estimation algorithm and the original grayscale frames. Furthermore, we calculate the PSNR values between the adjacent frames after applying the proposed global motion compensation algorithm. We compare the PSNR values obtained from these three algorithms to assess the level of image distortion. The results are presented in Figure 12 for visual comparison.

Figure 12 presents the PSNR value variations. p1 represents the PSNR value changes between the original adjacent frames, p2 represents the PSNR value changes between the compensated frames obtained by the traditional global motion method and the original grayscale frames, and p3 represents the PSNR value changes between the adjacent frames obtained by the proposed algorithm.

From the PSNR value variations in Figure 12a,b, it can be observed that the proposed algorithm achieves the highest PSNR values for the adjacent frames in both video sequences $V_{P}$ and $V_{Q}$. This indicates that the proposed algorithm yields the least distortion in the frames after global motion compensation. Therefore, it confirms the robustness and accuracy of the proposed algorithm.

The video sequence $V_{P}$ comprises 280 frames, while $V_{Q}$ consists of 210 frames. Figure 13 illustrates the runtime variation curves for each image in the frequency sequences $V_{P}$ and $V_{Q}$. Excluding the algorithm initialization time, the average processing time of the approach proposed in this paper for each image is 0.35 s.

Shao et al. [1] designed a lightweight parallel network (HRSiam) with a high spatial resolution to locate the small objects in satellite videos. Son et al. [39] proposed a real-time video deblurring framework (RVDMC) consisting of a lightweight multi-task unit that supports both video deblurring and motion compensation in an efficient way. Table 2 compares the running time of these two deep learning models and our proposed algorithm on one image. The runtime data for the deep learning models listed in Table 2 are based on PyTorch (PT) model executions and do not account for the impact of model acceleration techniques.

## 5. Conclusions

This paper proposes a global motion compensation algorithm for video object detection based on the affine inverse transformation model. The algorithm utilizes SURF in combination with the MSAC algorithm to obtain global motion parameters and further fits an affine transformation model. It estimates the background motion of the entire video sequence. Based on this, an inverse transformation model of affine transformation between adjacent frames is proposed. It is applied to globally compensate for the motion in the entire video sequence. The compensated frames are then outputted in a view created based on the first frame of the video sequence as a reference, completing the transformation from dynamic background to static background.

The effectiveness of the proposed algorithm is further demonstrated by evaluating the detected motion objects using frame differencing between adjacent frames and comparing the peak signal-to-noise ratio (PSNR) among different algorithms. However, due to the complexity and non-periodicity of background motion, the proposed algorithm may still introduce some noise in the detected objects. Further optimization and improvements are needed to address this issue.

The video processing results can be viewed at https://youtu.be/hMFII2rpc4s (accessed on 5 May 2023) and https://youtu.be/NAPBZkvBahg (accessed on 5 May 2023).

## Figures and Tables

**Figure 1 sensors-23-07750-f001:**
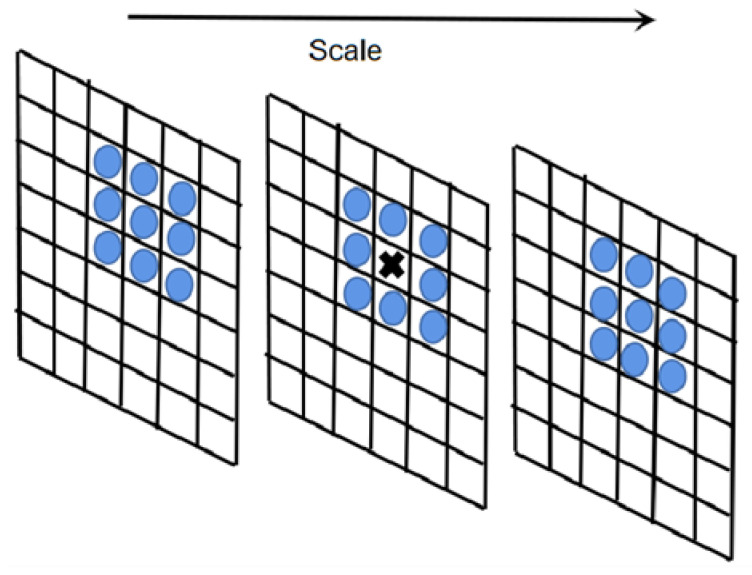
Schematic diagram of non-extremum suppression.

**Figure 2 sensors-23-07750-f002:**
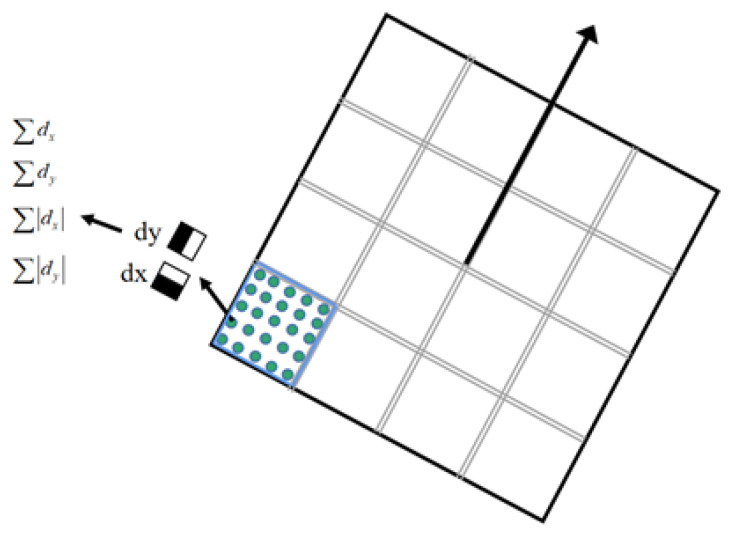
Feature point descriptor.

**Figure 3 sensors-23-07750-f003:**
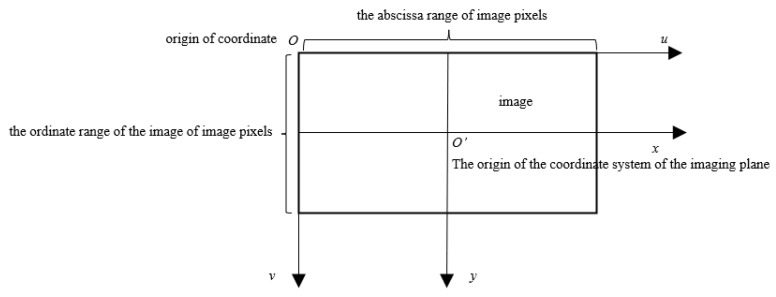
Image coordinate system and imaging plane coordinate system.

**Figure 4 sensors-23-07750-f004:**
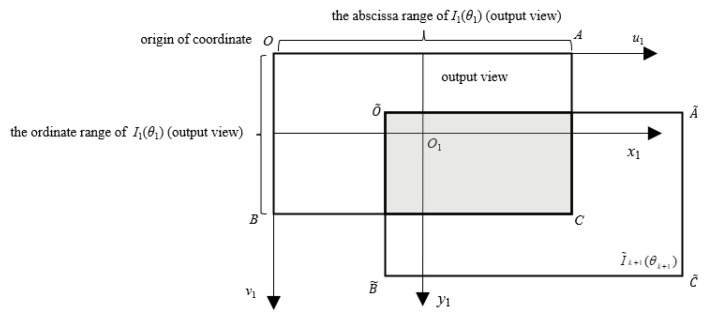
Output view.

**Figure 5 sensors-23-07750-f005:**
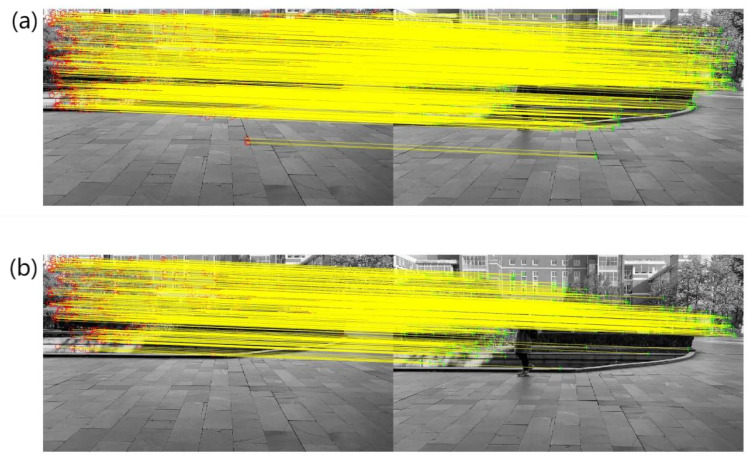
Matching results of $V_{P}$ key points and inliers. (**a**) $V_{P}$ key point matching; (**b**) $V_{P}$ inliers points matching.

**Figure 6 sensors-23-07750-f006:**
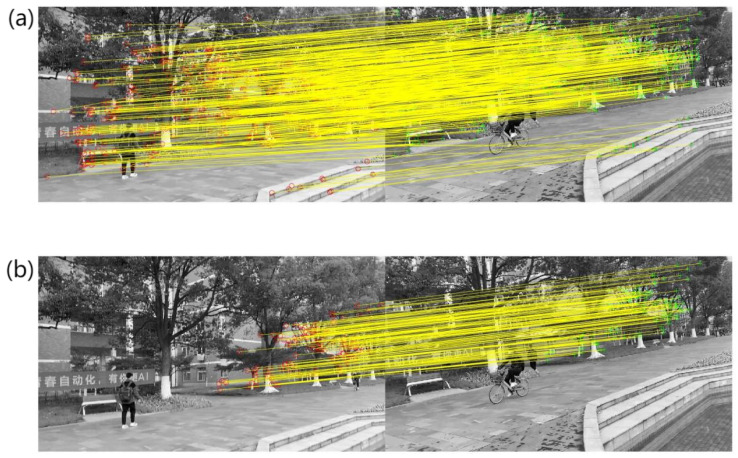
Matching results of $V_{Q}$ key points and inliers. (**a**) $V_{Q}$ key point matching; (**b**) $V_{Q}$ inliers points matching.

**Figure 7 sensors-23-07750-f007:**
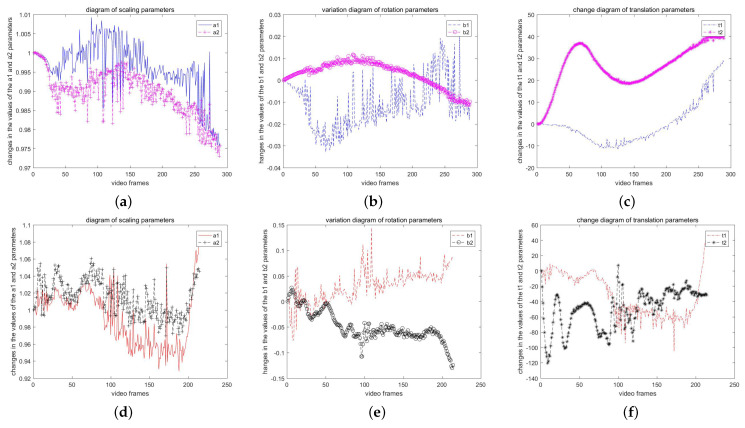
Variation diagram of background motion parameters. (**a**) change graph of VP scaling parameters; (**b**) variation diagram of VP rotation parameters; (**c**) variation diagram of $V_{Q}$ translation parameters; (**d**) change graph of $V_{Q}$ scaling parameters; (**e**) variation diagram of $V_{Q}$ rotation parameters; (**f**) variation diagram of $V_{Q}$ translation parameters.

**Figure 8 sensors-23-07750-f008:**
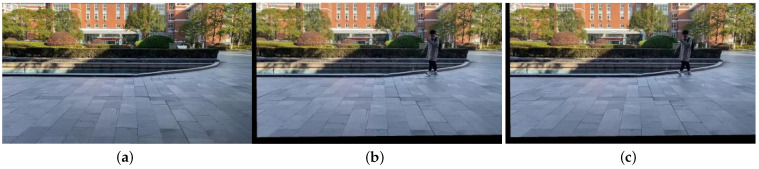
Three frames of $V_{P}^{\prime }$. (**a**) frame 1; (**b**) frame 125; (**c**) frame 126.

**Figure 9 sensors-23-07750-f009:**
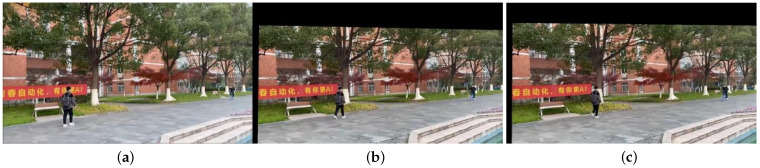
Three frames of $V_{Q}^{\prime }$. (**a**) frame 1; (**b**) frame 125; (**c**) frame 126.

**Figure 10 sensors-23-07750-f010:**
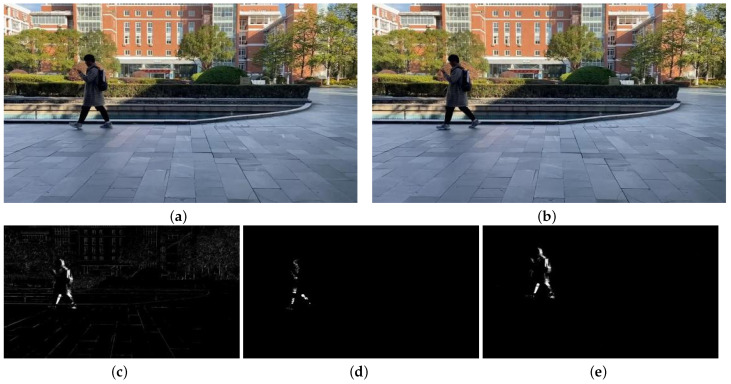
Comparison of $V_{P}$ sequence image detection results. (**a**) frame 223; (**b**) frame 224; (**c**) direct frame difference; (**d**) traditional algorithm; (**e**) our algorithm.

**Figure 11 sensors-23-07750-f011:**
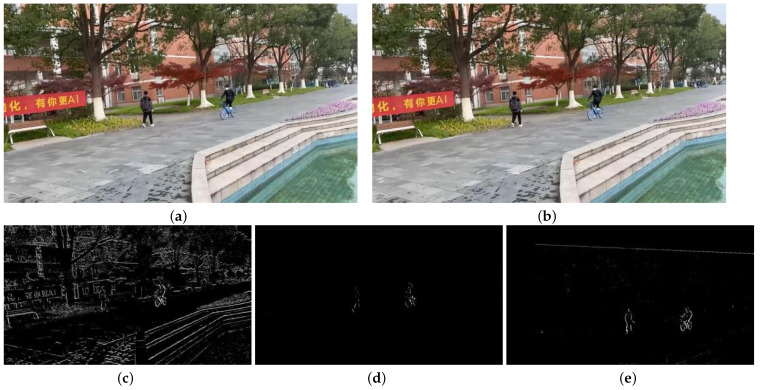
The results of the three algorithms. (**a**) frame 134; (**b**) frame135; (**c**) direct frame difference; (**d**) traditional algorithm; (**e**) our algorithm.

**Figure 12 sensors-23-07750-f012:**
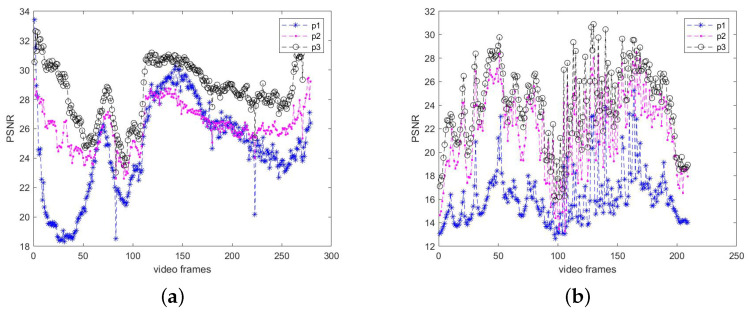
PSNR value change chart. (**a**) change diagram of PSNR value of video $V_{P}$; (**b**) change diagram of PSNR value of video $V_{Q}$.

**Figure 13 sensors-23-07750-f013:**
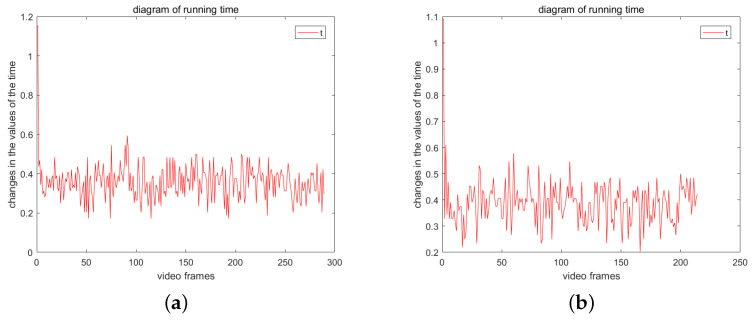
Runtime change chart. (**a**) change diagram of runtime of video $V_{P}$; (**b**) change diagram of runtime of video $V_{Q}$.

**Table 1 sensors-23-07750-t001:** Six parameter values for affine transformation between images.

Parameters	$a_{1}$	$a_{2}$	$b_{1}$	$b_{2}$	$t_{1}$	$t_{2}$
$V_{P}$	0.9953	0.9870	−0.0220	0.0023	−0.6703	54.1088
$V_{Q}$	0.9615	0.9935	0.0416	−0.0728	−63.5515	−61.2953

**Table 2 sensors-23-07750-t002:** The running time of each image is compared between the proposed algorithm and the deep learning algorithm on GPU and CPU.

Model	HRSiam	RVDMC	Ours
CPU Runtime	0.48	2.3	0.35

## Data Availability

Data openly available in a public repository. https://youtu.be/hMFII2rpc4s, accessed on 5 May 2023 and https://youtu.be/NAPBZkvBahg, accessed on 5 May 2023.

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
