# Peer review of "Video Global Motion Compensation Based on Affine Inverse Transform Model"

_sensors, 2023, doi:10.3390/s23187750_

Round 1
Reviewer 1 Report
The paper sounds overall timely and of possible interest for the readers of Sensors. The primary contribution of this paper is the proposal of a global motion compensation algorithm based on the affine inverse transformation model. This algorithm is utilized for detecting moving targets in video sequences with dynamic backgrounds. The research problem addressed is meaningful and demonstrates certain levels of innovation and practicality. Experimental results presented in the paper indicate that the algorithm effectively eliminates global motion in video sequences, improves the peak signal-to-noise ratio, and achieves satisfactory visual effects. These results provide support for the effectiveness and superiority of the algorithm. However, the paper also possesses certain limitations that require revision and supplementation in order to enhance its quality and credibility.
Please consider the following:
1) In the introduction, the author solely provides an exposition of the classical algorithms within the realm of global motion compensation, without delving into the contributions and comparative analysis of other researchers in the field in recent years. It is imperative that the author supplements the introduction by incorporating a comprehensive overview of the endeavors undertaken by scholars in this domain, while also emphasizing the innovative aspects of their own work.
2) The computational complexity and real-time performance of the global motion compensation algorithm are not sufficiently discussed in this paper. Specific time and space costs are not provided, and no comparison with other methods is presented. This aspect is crucial in video sequence processing since such sequences often possess high frame rates and resolutions. If the global motion compensation algorithm cannot handle video sequences quickly, it may adversely affect the efficiency and accuracy of target detection. Please ask the author to explain this issue.
3) In Figure 11(e), the frame difference image obtained by the proposed method exhibits a noticeable inclined dashed line. However, the author did not analyze this noise in the experiment. It is encouraged for the author to supplement the explanation regarding this observation.
4) In section 3.3, there are several instances where specific reference to certain images is not specified. It is kindly requested that the author rectify this omission.
The writing of the paper requires moderate improvement.
Author Response
We sincerely appreciate your advice. These views have certain reference value and help for our thesis. We have read these recommendations carefully and made changes and improvements. According to the instructions, we uploaded the reply instructions and the revised draft file. We want the new version to be smoother and clearer. Our responses to specific recommendations are contained in the annex.

Reviewer 2 Report
The paper proposes an estimation algorithm for affine transform between consecutive frames in video sequences for background motion compensation. The method uses the SURF algorithm to obtain matching points and the MSAC algorithm (a RANSAC variant) for outlier removal. The method is compared to a "traditional algorithm" in target matching.
The following issues should be addressed:
1. The parameters a_i and b_i in equation (19) (line 224, 225) are denoted as rotation and scaling parameters. In fact, the matrix can be composed of scaling (2 parameters), rotation (1 parameter) and shear (1 parameter). Denoting a_i as scaling and b_i as rotation parameters is misleading. This should be fixed, also around eqn (2).
2. It can be better described, how the six parameters a_i, b_i and t_i are estimated from the samples. Is this done by least squares fitting, contained in the MSAC iteration?
3. What ist the "traditional global motion estimation" compared to in figures 10 and 11? This algorithm should be summarized and referenced.
4. What are the limitations of the method? Knowing that the 2D affine transform approximates the process of projecting 3D-affinely trandformed scenes, the correct transform can be highly warped and not well described by only six parameters in the 2D affine model.
Minor issues:
eqn (5-7): use a star for convolution
eqn(9): where does the weight 0.9 come from? Why do we need the Hessian?
lines 261, 271: missing links to figures
All in all, an interesting approach.
Author Response
We really appreciate your advice. These views have certain reference value and help for our thesis. We have read these recommendations carefully and made changes and improvements. According to the instructions, we uploaded the reply instructions and the revised draft file. We want the new version to be smoother and clearer. Our responses to specific recommendations are contained in the annex.
